## Effect of balanced energy-protein supplementation during pregnancy and lactation on birth outcomes and infant growth in rural Burkina Faso: study protocol for a randomised controlled trial

Katrien Vanslambrouck [1], Brenda de Kok [1], Laeticia Celine Toe [1,2] Nathalie De Cock [1], Moctar Ouedraogo [3], Trenton Dailey-Chwalibóg [1] Giles Hanley-Cook [1], Rasmané Ganaba [3], Carl Lachat [1], Lieven Huybregts [4], Patrick Kolsteren [1]

► Prepublication history and additional materials for this paper is available online. To view these files, please visit the journal online (http://dx.doi.org/10.1136/bmjopen-2020-038393).

For numbered affiliations see end of article.

**Correspondence to**
Katrien Vanslambrouck;
katrien.vanslambrouck@ugent.be and
Prof Carl Lachat;
carl.lachat@ugent.be

## ABSTRACT

**Introduction** Adequate nutrition during pregnancy is crucial to both mother and child. Maternal malnutrition can be the cause of stillbirth or lead to poor birth outcomes such as preterm delivery and small-for-gestational-age newborns. There is a probable positive effect of providing pregnant women a balanced energy-protein (BEP) food supplement, but more evidence is needed. The MIcronutriments pour la SAnté de la Mère et de l'Enfant (MISAME) III project aims to improve birth outcomes and infant growth by testing a BEP supplement during pregnancy and lactation in rural Burkina Faso. This paper describes the study protocol.

**Methods and analysis** MISAME-III is a four-arm individually randomised efficacy trial implemented in six rural health centre catchments areas in the district of Houndé. Eligible pregnant women, aged between 15 and 40 years old and living in the study areas, will be enrolled. Women will be randomly assigned to one of the four study groups: (1) prenatal intervention only, (2) postnatal intervention only, (3) prenatal and postnatal intervention or (4) no prenatal or postnatal intervention. The intervention group will receive the BEP supplement and iron/folic acid (IFA) tablets, while the control group will only receive the IFA tablets following the national health protocol. Consumption will be supervised by trained village women on a daily basis by means of home visits. The primary outcomes are small-for-gestational age at birth and length-for-age z-score at 6 months of age. Secondary outcomes will be measured at birth and during the first 6 months of the infants' life. Women will be enrolled from October 2019 until the total sample size is reached.

**Ethics and dissemination** MISAME-III has been reviewed and approved by the University Hospital of Ghent and the ethics committee of Centre Muraz, Burkina Faso. Informed consent will be obtained. Results will be published in relevant journals and shared with other researchers and public health institutions.

**Trial registration number** NCT03533712.

### Strengths and limitations of this study

► This trial will help to fill the evidence gap on the effect of balanced energy-protein (BEP) supplements in pregnant and lactating women on birth outcomes and infant growth.
► Formative research to select the most suitable BEP supplement ensured that the selected BEP is well accepted by the study population.
► The daily intake of BEP supplements and iron/folic acid tablets during pregnancy and lactation will be directly observed by study workers.
► This study will assess the impact of factorial combinations of prenatal and postnatal BEP on child growth to elucidate the relative importance of BEP during pregnancy and/or early lactation.
► Blinding of study participants and staff members will not be possible, as the supplements are identifiable.

## INTRODUCTION

Pregnancy is a challenging period in the life of many women in low-income and middle-income countries (LMICs). Maternal mortality remains high, and many neonates suffer from premature delivery and/or intra-uterine growth retardation, both in length and in weight accumulation.[1] An indicator to measure neonatal growth is small-for-gestational age (SGA). SGA is defined as a birth weight below the 10th percentile of a standard optimal reference population for a given gestational age and sex.[2] SGA is often caused by growth restriction in the womb and has been associated with neonatal and postneonatal mortality.[2] It has also been linked to an increased risk of morbidity later

in life, especially non-communicable diseases.[3] SGA affected 23.3 million term children in LMICs in 2012.[4] Adequate nutrition during pregnancy is crucial for optimal maternal and newborn health,[5 6] and maternal malnutrition has been associated with fetal growth restriction.[7] An adequate dietary balance is necessary to ensure sufficient energy intake for adequate growth of the fetus.[8] Unfortunately, maternal undernutrition remains a public health challenge in regions across sub-Saharan Africa and Asia.[9 10]

Several types of food supplements have been developed and evaluated over the past years. A positive effect of multiple micronutrient supplements (MMS) during pregnancy on birth outcomes has been found in previous studies.[11] Keats *et al*[11] concluded in their review that MMS during pregnancy gave a probable reduction in SGA and preterm births and can thus be used for future guidance. According to a multicountry randomised controlled trial (RCT) done in LMICs, a positive effect of lipid-based nutrient supplements on fetal growth-related birth outcomes can be seen when starting supplementation before conception or during the first trimester.[12] Moreover, the latest evidence indicates a possible positive effect of providing pregnant women a balanced energyprotein (BEP) food supplement.[5 13–15] In line with that evidence, the 2016 WHO's antenatal care guidelines state that pregnant women in undernourished populations should receive, depending on the context, BEP supplements to reduce the risk of stillbirth and SGA.[6] Researchers, however, still highlight the limited amount of evidence and a need to evaluate the effect of balanced supplements on birth outcomes, such as SGA.[5 13] Experimental trials of high-quality and large sample sizes, especially in undernourished pregnant women, are thus needed.[13] Following this recommendation, compositional guidance for a ready-to-use food supplement for pregnant women was developed by the Bill and Melinda Gates Foundation (BMGF) in 2016.[16]

Two previous projects, MIcronutriments pour la SAnté de la Mère et de l'Enfant (MISAME) I and II, conducted in Burkina Faso, investigated the effect of supplementation during pregnancy and the effect on birth outcomes in infants.[17 18] MISAME-I compared the effect of the UNICEF/WHO/UNU international multiple micronutrient preparation (UNIMMAP) with the effect of iron/folic acid (IFA) alone on fetal growth in a double-blind RCT and concluded that UNIMMAP modestly but significantly increased fetal growth.[17] The second study (MISAME-II) assessed the effect of a lipid-based nutrient supplement fortified with UNIMMAP compared with a UNIMMAP tablet during pregnancy in an open-label, individually randomised controlled trial on birth anthropometry. It was found that combining energy with micronutrients during the prenatal phase led to larger birth lengths.[18] Since MISAME-II used the UNIMMAP as a control, a complete assessment of the impact of the fortified lipid-based UNIMMAP was not possible. Therefore, MISAME-III will study the effect of a BEP

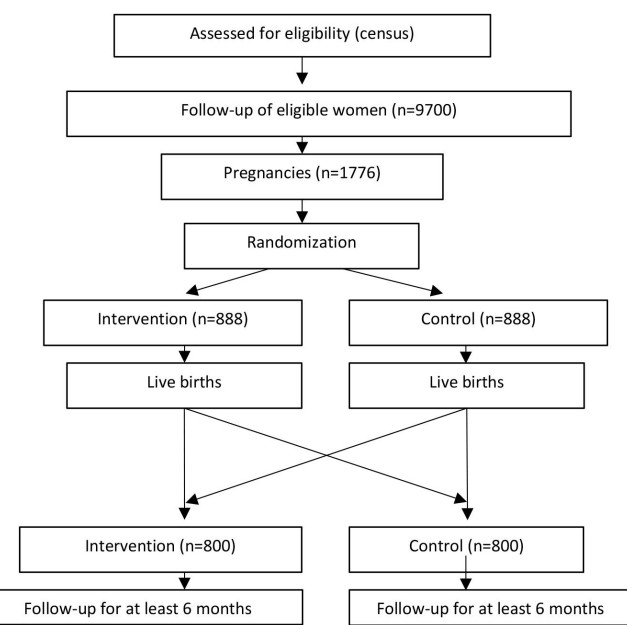

**Figure 1** Study design of the randomised controlled trial.

supplementation, compared with a control group, and extend the supplementation postnatally to investigate the net contribution of prenatal and postnatal BEP supplementation on child linear growth up to 6 months of age.

In summary, the MISAME-III study hypothesises that: (1) providing women with a BEP supplement during pregnancy will decrease the incidence of SGA compared with the control group; and (2) providing them with a BEP supplement during the postnatal period will increase children's length by the age of 6 months compared with the control group.

## METHODS
This protocol has been developed in accordance with the Standard Protocol Items: Recommendations for Interventional Trials guidelines (online supplemental file 1).

### MISAME-III study design
The MISAME-III project is an individually randomised 2×2 factorial efficacy trial aiming to improve birth outcomes and infant growth in rural Burkina Faso by testing a BEP supplement during pregnancy and lactation (figure 1). At inclusion, pregnant women will be individually and randomly allocated to a prenatal intervention or control group and a postnatal intervention or control group. The intervention group will receive a daily BEP supplement to be consumed under supervision for the duration of pregnancy/lactation. Both intervention and control groups will receive the standard IFA tablet through the national antenatal care. In addition to the main trial, we propose a number of substudies to test specific hypotheses in a subsample of pregnant/lactating women and children. MISAME-III began with a formative study to identify the preferred product type for the provision of a fortified BEP supplement during the RCT.

## Study setting

The study will be conducted in the district of Houndé in Burkina Faso, a landlocked country situated in West-Africa; similar to the previous MISAME studies, Burkina Faso has an infant mortality rate of 53 per 1.000 live births,[19] with an estimated Low Birth Weight (LBW) prevalence at 14% in 2013.[20] The prevalence of SGA has been estimated to be between 32.2% and 41.6% in the district of Houndé.[17] The Demographic and Health Survey of 2010 reported that 16% of women had a body mass index (BMI) below 18.5 kg/m², which indicates the presence of chronic energy deficiencies in the zone.[21] The highest prevalence can be found in the Eastern region, where 31% of women have a BMI lower than 18.5 kg/m², that is, low BMI.[22] Moreover, in particular, adolescent Burkinabèe girls between the age of 15 and 19 years have a low BMI, with an estimated prevalence of 23%.[23] Micronutrient deficiencies also remain a major problem in both infants and women of reproductive age in the country.[23 24]

The climate of the country is Sudano-Sahelian, with a dry season from October to March/April and a rainy season from May until September/October. The diet is essentially cereal based[25] with maize as the main staple food.[26]

MISAME-III will be conducted in the same health district where the two previous MISAME studies were organised. The study villages are concentrated around six health centres, which are within an accessible range from the district hospital. A list of all study sites can be found on: www.misame3.ugent.be

## Study population and recruitment

Women living in the study villages and aged between 15 and 40 years will be identified through a census. The villages were selected based on their accessibility and distance to the nearest health centre, number of facility-based deliveries per year and their agricultural model as some households tend to reside on their fields during the harvest season. Trained village women (femme accompagnante (FA)), selected in collaboration with the community leaders, will visit the households every 5 weeks to ask about women's menstruation. In case of amenorrhoea, women will be sent to the nearest health centre for a pregnancy test and a first antenatal consultation by our project midwife when tested positive. An ultrasound examination will be completed soon after inclusion by the project medical doctor to assess gestational age. A baseline interview will also be done by the project interviewers to assess the household members' characteristics, household properties, water sanitation and hygiene (WASH) and household food security.

Inclusion criteria:
► Women between 15 and 40 years old at study inclusion.
► Confirmed pregnancy by a pregnancy test and ultrasound.
► Women who signed the informed consent form.

Exclusion criteria:
► Pregnancies >20 weeks of gestational age.
► Women planning on leaving the area during their pregnancy.
► Women planning on delivering outside the study area.
► Women who are allergic to peanuts.
► Women with multifetal gestations (exclusion from analysis).

FAs will be informed by the project midwife when a participant has been included. FAs will visit pregnant women on a daily basis to distribute the BEP supplement and/or IFA tablet and to supervise consumption. During the postnatal period, FAs will distribute the supplements and IFA tablets to the intervention group on a daily basis until 6 weeks after birth. From that moment onwards, they will receive a week's worth of BEP supplements. The postnatal control group will receive the IFA tablets on a daily basis during the first 6 weeks after birth, and participants will thereafter be visited once a week (without any supplementation) to minimise the effect of home visits. The FA will inform women on the supplement's function, the importance of antenatal visits during pregnancy, maintaining a healthy diversified diet, the importance of delivering at a health facility, the importance of exclusive breast feeding and the introduction of complementary foods at the age of 6 months. Throughout the study, the FAs will be supervised by project interviewers. Supervision visits will be conducted using Lot Quality Assurance Sampling schemes and empty sachet counts to ensure that study participant are visited according to the project protocol.

## Manufacturing of 12 fortified BEP supplements and the formative study

Twelve fortified BEP supplements were pretested before the start of the RCT during a formative research phase. Several food manufacturing companies were invited to produce ready-to-use BEP supplements following the compositional guidelines proposed during an expert meeting hosted by the BMGF[16] in September 2016 (table 1). The BEP supplements had to be: (1) ready to consume, (2) not need a cold chain and (3) microbiologically stable.

Seven out of 12 supplements were characterised as sweet and five as savoury. Products were produced in different forms, including a biscuit, pillow, wafer, bar, paste, instant drink and soup.

In a first screening step of the formative study, the two most preferred BEP supplements were identified by using a combined evaluation approach consisting of a single meal test, sensory evaluation and focus group discussions, in a convenience sample of 40 pregnant women. In a next step, we compared the acceptability of the two preselected BEP supplements using a 10-week home-feeding study, with 80 pregnant women, to select to most preferred product for the RCT. We refer to both papers for detailed information.[27 28]

## Study intervention

At inclusion, pregnant women will be randomly allocated to four different study groups: (1) prenatal intervention

**Table 1** The compositional guidelines for macronutrients and micronutrients

| Nutrition component | Target per serving |
|---|---|
| Total energy | 250–500 kcal per serving. |
| Fat content | 10%–60% of energy intake. |
| Protein content | 16 g (14–18 g) with a digestible indispensable amino acid score of ≥0.9. |
| Carbohydrates | Between 45 g and 32 g per 100 g (added sucrose between 20 g and 10 g per 100 g). |
| Trans fats | <1% of energy intake. |
| Fatty acid | (Optional): min of 1.3 g of n-3 or 300 mg docosahexaenoic acid (DHA)+eicosapentaenoic acid (of which 200 mg DHA) to achieve a healthy n-6: ratio of the supplement of 5:1. |
| Micronutrients | Vitamin A, D, E, K, B1 (thiamin), B2 (riboflavin), B3 (niacin), B6 (pyridoxine), B9 (folate), B12 and C; minerals: iron, zinc, iodine, calcium, phosphorous, copper and selenium. |
| Optionally | Pantothenic acid, manganese, potassium, biotin and choline will be included. |
| | The final composition of the product will be determined by the selected product as the manufacturing process will influence the macronutrient composition. |

only, (2) postnatal intervention only, (3) prenatal and postnatal intervention or (4) no prenatal and no postnatal intervention. Prenatal and postnatal supplementation will start right after inclusion and birth, respectively. The prenatal intervention group will receive the BEP supplement and IFA, while the control group will receive IFA tablets alone. The IFA tablets contain 65 mg iron and 0.4 mg folic acid. The postnatal intervention group will receive the BEP supplement for 6 months in combination with the IFA tablets for 6 weeks following the national protocol of Burkina Faso; the control group will receive IFA alone for 6 weeks.

### Allocation/randomisation

We will apply a stratified permuted block randomisation schedule to allocate women to the prenatal intervention or control group and in a next step to allocate women to a postnatal intervention and control group. Per health centre (ie, stratum), women will be individually randomly in permuted blocks of 8 so that, per block, equal numbers are obtained in the prenatal intervention (n=4) and control (n=4) group and equal numbers are also obtained in the postnatal intervention (n=4) and control (n=4) group. The double random sequence will be generated before the start of the study using Stata V.15.1 (Statacorp, Texas, USA) by an external research analyst. The allocation group will be coded with two letters (A or B for the prenatal and Y or Z for the postnatal study group) and

placed in a sequentially numbered sealed opaque envelopes by project employees, not in direct contact with participants. At study enrolment, the project midwife will draw the next sealed envelope and allocate the participant to the study group defined by the letter code in the envelope. Blinding of participants and community-based project workers will not be possible since the supplements are identifiable. Field staff responsible for measuring primary and secondary study outcomes are not directly involved in the daily supplementation of the study participants and can therefore be considered to be partially blinded.

### Outcomes

#### Primary outcomes of the RCT

The trial has two primary study outcomes that will be used to assess the impact of the prenatal and the postnatal intervention, respectively:

► Incidence of SGA, defined as birth weight <10th centile of the Intergrowth 21st reference.[29]
► Length-for-age z-score (LAZ) calculated using the WHO 2006 growth reference at 6 months of age.[30]

#### Secondary outcomes of the RCT

A list of the trial's secondary outcomes can be found in table 2.

Birth weight measurements will be defined using the Intergrowth 21st reference,[29] and child anthropometry will be defined using the Child Growth Standards developed by the WHO.[30]

#### Outcomes of the substudies

Substudy 1: impact of the intervention on neonatal and maternal body composition 2–3 weeks after delivery.

Body composition will be determined in mother–child dyads by deuterium dilution and analysis of saliva by a Fourier Transform Infrared reader (Agilent FTIR 4500 series). The substudy will also assess if early gestation maternal BMI (defined as body weight in kilograms divided by the square of height in metres) modifies the intervention's effect on neonatal body composition.

Substudy 2: impact of the intervention on dietary intake.

A dietary assessment study will be conducted, using a 24-hour dietary recall in a subsample of women. This substudy will enable us to assess possible substitution of the prenatal diet by the BEP supplement.

Substudy 3: impact of the intervention on breastmilk.

Breastmilk samples will be taken in the four study groups to compare the composition and to analyse the interaction between the supplementation periods.

### Sample size

With an SGA prevalence of 32% and an anticipated decrease of 7%, a sample of 652 subjects per prenatal arm is required with $\alpha=0.05$ and $\beta=0.2$.[26] To accommodate for possible losses, the number of subjects per arm was increased to 888 (total subjects: 1776). Possible losses are based on previous MISAME studies where the prevalence

**Table 2** Secondary outcomes of the RCT on maternal, newborn and child level

| Maternal outcomes | Newborn | Child |
|---|---|---|
| Total and trimester-specific prenatal weight gain and gestational weight change. | Birth weight (measured within 72 hours after birth). | Weight-for-age Z-score at 6 months of age (WAZ) (and 9 and 12 months on a subsample). |
| Probable and possible maternal postnatal depression at 2 and 6 months of child age. | Birth length (measured within 72 hours after birth). | Weight-for-length Z-score at 6 months of age (WLZ) (and 9 and 12 months on a subsample). |
| Maternal anaemia at the third antenatal consultation. | Ponderal or Rohrer's index at birth. | Stunting at 6 months of age. |
| Women's mean and minimum dietary diversity score (measured twice weekly). | Gestational age at delivery. | Wasting at 6 months of age. |
| | Large-for-gestational age. | Underweight at 6 months of age. |
| | Chest circumference (measured within 72 hours after birth). | Duration of exclusive breastfeeding during the first 6 months of age. |
| | Head circumference (measured within 72 hours after birth). | Incidence of child wasting over first 6 months of life. |
| | Arm circumference (measured within 72 hours after birth). | Weight gain over first 6 months of life. |
| | Incidence of preterm birth. | Child mortality (between birth and 6 months of age). |
| | Fetal loss. | Monthly change in LAZ over first 6 months of life. |
| | Stillbirths. | Monthly change in WLZ over first 6 months of life. |
| | | Monthly change in WAZ over first 6 months of life. |
| | | Monthly change in head circumference over first 6 months of life. |
| | | Child morbidity symptoms over first 6 months of life. |
| | | Anaemia at 6 months of age. |
| | | Haemoglobin concentration at 6 months of age. |

LAZ, length-for-age z-score; RCT, randomised controlled trial.

was ~26 % due to a combination of miscarriage, stillbirths, multifetal pregnancies, outmigrants, maternal deaths and incomplete data.[18] For the analysis of an effect of the postnatal intervention on LAZ at 6 months of age, the minimally detectable effect depends on the presence or absence of an interaction effect between the prenatal and postnatal intervention. In the absence of a statistically significant interaction between prenatal and postnatal intervention, at sample size of 588 children per postnatal study arm would allow us to detect a difference of 0.18 Z-score (SD=1.1) based on a cross-sectional survey conducted in the Gourcy health district in Burkina Faso,[31] between study arms with α=0.05 and β=0.20. This implies that if ~1400 singleton live births are available, we allow for a maximum loss to follow-up of 16%. In the presence of a statistically significant interaction between prenatal and postnatal intervention, a total sample size of 1176 represents 294 children per factorial combination of the prenatal and postnatal study group (four groups in total). A subgroup size of 294 would allow us to detect

a difference in LAZ at 6 months of age of 0.28 assuming an SD of 1.1, α=0.025 (Bonferroni correction for two primary endpoints analyses) and β=0.20.

### Data collection

#### Anthropometric and clinical procedures

At enrolment, anthropometric measurements from all women will be taken. Gestational age will be determined during an ultrasound consultation by measuring crown-rump length (7–13 weeks) or by calculating the mean of three to four measurements: biparietal diameter, head circumference, abdominal circumference and femur length (12–26 weeks).[32]

During pregnancy, clinical follow-up will consist of antenatal visits following the national guidelines.

At birth, anthropometric measurements of all neonates will be assessed in duplicate within the first 72 hours of life (in practice, the aim is to measure within the 24 hours of life). After birth, mother and child will visit the healthcare centres monthly for a follow-up on clinical,

**Table 3** Participant timeline schedule of enrolment, interventions, assessment and visits

| | Enrolment | | Allocation | Postallocation | | | | After birth | | | | | | | |
| | | | | Pregnancy and birth | | | | | | | | | | | |
| | Start | 5-weekly visits | ANC 1 | Household visit | Ultrasound | ANC 2, 3 and 4 | Birth | Month 1 | Month 2 | Month 3 | Month 4 | Month 5 | Month 6 | Month 9 | Month 12 |
|---|---|---|---|---|---|---|---|---|---|---|---|---|---|---|---|
| **Enrolment** | | | | | | | | | | | | | | | |
| Census | × | | | | | | | | | | | | | | |
| Pregnancy identification | | × | | | | | | | | | | | | | |
| Pregnancy confirmatory test | | | × | | | | | | | | | | | | |
| Informed consent and allocation to study group | | | × | | | | | | | | | | | | |
| **Study groups** | | | | | | | | | | | | | | | |
| Prenatal BEP+IFA | | | ×——————————————× | | | | | | | | | | | | |
| Prenatal IFA | | | ×——————————————× | | | | | | | | | | | | |
| Postnatal BEP+IFA* | | | | | | | ×——————————————————————× | | | | | | | | |
| Postnatal IFA | | | | | | | ×————————× | | | | | | | | |
| **Assessments** | | | | | | | | | | | | | | | |
| **Mothers** | | | | | | | | | | | | | | | |
| Baseline questionnaire | | | × | | | | | | | | | | | | |
| Gestational age determination | | | | × | | | | | | | | | | | |
| Skinfold measurements | | | | × | | | | | | | | | | | |
| Weight (kg) and arm circumference (mm) †/‡ | | × | | | | × | × | × | × | × | × | × | × | × | × |
| Height (cm) | | | × | | | | | | | | | | | | |
| Haemoglobin (g/dL) | | | × | | | × (ANC3) | | | | | | | | | |
| Women's Dietary Diversity Score (twice weekly) | | | × | | | | | | | | | | | | |
| Maternal depression | | | × | | | | | | × | | | | × | | |
| **Infants** | | | | | | | | | | | | | | | |
| Birth weight (kg) | | | | | | | × | | | | | | | | |
| Birth length (cm) | | | | | | | × | | | | | | | | |
| Head circumference (mm) | | | | | | | × | × | × | × | × | × | × | | |
| Chest circumference (mm) | | | | | | | × | | | | | | | | |
| Arm circumference (mm) | | | | | | | × | × | × | × | × | × | × | × | × |
| Morbidity | | | | | | | | × | × | × | × | × | × | | |
| Mortality | | | | | | | × | × | × | × | × | × | × | | |
| Weight (kg) and height (cm) | | | | | | | × | × | × | × | × | × | × | × | × |

Continued

**Table 3** Continued

| | Enrolment | | Allocation | Postallocation | | | | | | | | | | | | |
| | | | | Pregnancy and birth | | | | After birth | | | | | | | |
| | Start | 5-weekly visits | ANC 1 | Household visit | Ultrasound | ANC 2, 3 and 4 | Birth | Month 1 | Month 2 | Month 3 | Month 4 | Month 5 | Month 6 | Month 9 | Month 12 |
|---|---|---|---|---|---|---|---|---|---|---|---|---|---|---|---|
| Breastfeeding practices | | | | | | | | × | × | × | × | × | × | | |
| Haemoglobin (g/dL) | | | | | | | | | | | | | × | | |

*The IFA tablets will be given during the first 6 weeks after birth in the postnatal intervention group, according to the national health protocol.
†Only maternal weight will be taken at birth.
‡In a subsample at months 9 and 12.
BEP, balanced energy–protein; IFA, iron/folic acid.

anthropometric and child morbidity measures (signs including fever, vomiting, diarrhoea, cough, difficulty breathing and running nose). A subsample will be measured at the healthcare centres or at home by the project interviewers to collect postnatal data at months 9 and 12.

Haemoglobin concentrations will be measured in women at enrolment and during the third antenatal care visit. This will be conducted at 6 months of age among children.

### Baseline questionnaires

Prenatal and postnatal maternal depression will be assessed using the standardised Edinburgh Postnatal Depression Scale questionnaire consisting of 10 questions.[33] Project midwives will be trained for this, and the questionnaire will be asked at inclusion and at months 2 and 6 after birth. Socioeconomic and demographic information from all participants will be collected once included. Trained project interviewers will ask questions on household members' characteristics, household properties, WASH environment and household food security.[34] The women's dietary diversity score will be measured in all participating women by the FA during the home visits. This will be enumerated twice a week per participant using the Women's Dietary Diversity Score with 11 food groups.[35]

Table 3 shows the overview of the time schedule and measurements of the trial.

Quality of all study data will be insured by a thorough training of all field staff. Procedures to handle data collection tools (questionnaires, anthropometric and clinical measurement material, and laboratory procedures) will be pretested in the field during a dry-run of ±3 months. Anthropometric measurement standardisations of the field staff will be repeated bimonthly throughout the trial. Anthropometric measurements will be taken in duplicate. Newborns will be measured within 72 hours after birth (preferably within the 24 hours), and all weighing scales and HemoCue 201+ devices will undergo weekly quality control. A WhatsApp group will be set up where problems can be communicated and solved quickly.

All data collection forms of the trial can be found on: www.misame3.ugent.be

Women will be designated as lost to follow-up if they move from the study area or withdraw their participation. Reasons for discontinuation will be recorded.

Women will be enrolled in the study from October 2019 until the total sample size has been reached.

### Data management and analysis

FAs will use smartphones with computer-assisted person interviewing programmed in CSPRO (version 7.3.1) to collect data during household visits. The study data collected by the project medical doctor, project midwives and interviewers will be done by Survey Solutions data entry software (V.19.12.6) on tablets. This data will be uploaded to a central server on a weekly basis. All questionnaires were programmed and have been tested on the Survey Solutions Designer website and include validation codes to promote the quality of the data entry in the field. Assignments will be sent once a week to the tablets of the field team, and preloaded data collected at an earlier contact moment will be used to lower the amount of incorrect data. Paper forms will also be available on the field as a backup.

Further data quality checks will be conducted in Stata V.14.2 (Statacorp, Texas, USA). Missing or inconsistent data outliers will be sent back to the field for revision.

### Statistical analysis

We refer to the Statistical Analysis Plan of the trial '*Statistical analysis plan: Impact of a prenatal and postnatal balanced energy-protein supplement on birth size and postnatal child growth in Burkina Faso*' published on: www.misame3.ugent.be

### Data monitoring

#### Data monitoring and auditing

The Data and Safety Monitoring Board is an independent multidisciplinary group whose members are not involved in the trial. The board consists of a Belgian endocrinologist, a Belgian paediatrician, a Burkinabè paediatrician, a Belgian gynaecologist and a Belgian ethicist.

## Serious adverse events (SAEs)

FAs will be trained to recognise health issues and will actively refer those participants to see the project midwife in the primary health facilities or Centre de Santé et Promotion Sociale (CSPS) in the event they occur. All SAEs will be recorded on a case-by-case basis, and verbal autopsies will be conducted for maternal, neonatal and infant deaths by the field medical doctor.

## ETHICS AND DISSEMINATION
### Ethics approval and consent to participate

MISAME-III has been reviewed and approved by the University Hospital of Ghent University (B670201734334) and the Burkinabe ethics (N°2018–22/MS/SG/CM/CEI) committee. Important protocol changes will be noted on ClinicalTrials.gov. When eligible women meet the inclusion criteria, project midwives will explain the background and procedure of the complete trial. Written informed consent or assent will be asked from the participating women. In case of illiteracy, a thumb print will be asked and witnessed by the recruiting investigator and one extra witness. Participants will be told that all data collected during the trial is confidential and that they are allowed to withdraw at any time. A copy of the informed consent and assent can be found on www.misame3.ugent.be and as supplementary file (online supplemental file 2).

### Patient and public involvement

MISAME-III has been well accepted by the community, because of the previous positive experiences they had with the MISAME-I and II studies. Through the formative study, women were involved in the choice of BEP supplement. Workshops will be planned at the end of the study in order to communicate the study results to the community.

### Ancillary care

The MISAME-III project will pay for ancillary care when participants have health issues and in case the costs are not covered by the national healthcare programme. Participants suffering harm due to their trial participation will be covered.

### Confidentiality

A data management plan has been put in place to address concerns regarding the General Data Protection Regulation rules. During the trial, the data files containing personal identifying information will be stored on the Survey Solutions server. Only the principal investigators and the project coordinators will be able to access those files.

### Dissemination plan

On completion of the trial, all anonymised study data will be available on request. Final results will be communicated to the participants, the Burkinabè Ministry of health, the field staff, the BMGF, Ghent University researchers and students, AFRICSanté,

healthcare professionals and other relevant international public institutions. Papers on the study results will be published in peer-reviewed journals and will be available on the project website. All investigators contributing to the realisation of the project and publication of results will be included as an author. Other contributors such as the participants, FA and field staff members will be mentioned in the acknowledgements.

MISAME-III has been well accepted by the community, because of the previous positive experiences they had with the MISAME-I and II studies. Through the formative study, women were involved in the choice of BEP supplement. Workshops will be planned at the end of the study in order to communicate the study results to the community.

## DISCUSSION

In this paper, the protocol of an individually randomised four-arm efficacy trial in rural Burkina Faso has been described in which pregnant and lactating women in the intervention group will receive a BEP supplement together with IFA tablets. The control group will only receive the standard IFA treatment.

The key features of the present trial are, first, the inclusion of a formative study for a better understanding of which type of supplement is preferred, what taste is most acceptable and which factors affect adherence in the study population. Second, the supplementation will be given during pregnancy and during the first 6 months after birth. This will give us the opportunity to assess the specific value of postnatal supplementation on several outcomes. Third, the observed daily intake of intervention and control supplements is a key feature to ensure compliance and to avoid sharing of the supplements with other household members. Fourth, MISAME-III has the advantage of being the third trial of its kind in the study area. This presents an opportunity to anticipate the issues that arose in previous trials. For instance, women in specific villages tended to leave their homes for a longer period to go work on the fields outside the village. This posed problems in the continuation of the supplementation in the past and will be taken into consideration during MISAME-III. Fourth, four substudies are nested in the main trial that will provide insight into the mechanism by which prenatal BEP supplementation affects birth and infant outcomes. And last, similar studies are being conducted in other countries, allowing for comparison between results from different contexts.

The MISAME-III study will provide evidence on the impact of BEP supplements on birth and infant size using a rigorous study design. The study results will further strengthen and refine WHO's recommendation on the use of context-specific BEP supplementation during pregnancy and lactation.

**Author affiliations**
[1]Department of Food Technology, Safety and Health, Faculty of Bioscience Engineering, Ghent University, Gent, Belgium
[2]Institut de Recherche en Sciences de la Sante, Bobo-Dioulasso, Burkina Faso
[3]AFRICSanté, Bobo Dioulasso, Burkina Faso
[4]Poverty, Health and Nutrition Division, International Food Policy Research Institute, Washington, DC, USA

**Acknowledgements** The MIcronutriments pour la SAnté de la Mère et de l'Enfant (MISAME) Study Group would like to thank the pregnant women and their families for the time spent in this study. We would like to acknowledge the staff of AFRICSanté; including Henri Somé for his support on the computer-assisted person interviewing software, the field medical Dr Anderson Compaoré, the data collectors (midwives, interviewers and femmes accompagnantes), Dr Alain Hein for the standardisation exercises and Dr Hermann Lanou for his support during the training. We would like to thank Dr Sheila Isanaka for her collaboration during the formative research. Our private sector partner Nutriset (France) is acknowledged for donating the balanced energy-protein supplements.

**Contributors** KV wrote the manuscript; PK, LH, CL, NDC, LCT, KV, BdK, TD-C and GHC designed the study and the protocol; PK, LH, LCT, KV and BdK designed the study material tools; LCT, LH, KV, BdK, MO and RG trained the field data collectors; TD-C and GHC critically reviewed and revised the manuscript; all authors contributed substantially to the manuscript and approved the final version.

**Funding** This work was funded by the Bill and Melinda Gates Foundation. Grant number OPP1175213. The trial is registered on Clinical Trials.gov (identifier: NCT03533712) prior to recruitment. Enrolment started in October 2019.

**Competing interests** None declared.

**Patient consent for publication** Not required.

**Provenance and peer review** Not commissioned; externally peer reviewed.

**ORCID iDs**
Katrien Vanslambrouck http://orcid.org/0000-0003-1746-7056
Brenda de Kok http://orcid.org/0000-0002-5267-327X
Laeticia Celine Toe http://orcid.org/0000-0002-4615-5388
Nathalie De Cock http://orcid.org/0000-0002-0053-0269
Moctar Ouedraogo http://orcid.org/0000-0002-1521-0532
Trenton Dailey-Chwalibóg http://orcid.org/0000-0002-8204-4925
Giles Hanley-Cook http://orcid.org/0000-0001-9907-594X
Rasmané Ganaba http://orcid.org/0000-0001-7401-9546
Carl Lachat http://orcid.org/0000-0002-1389-8855
Lieven Huybregts http://orcid.org/0000-0002-3068-2853
Patrick Kolsteren http://orcid.org/0000-0002-0504-2205

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
