## [Reviewer comments · BMJ Open]

ARTICLE DETAILS

TITLE (PROVISIONAL)	THE EFFECT OF BALANCED ENERGY-PROTEIN SUPPLEMENTATION DURING PREGNANCY AND LACTATION ON BIRTH OUTCOMES AND INFANT GROWTH IN RURAL BURKINA FASO : STUDY PROTOCOL FOR A RANDOMIZED CONTROLLED TRIAL
AUTHORS	Vanslambrouck, Katrien; de Kok, Brenda; Toe, Laetitia Celine; De Cock, Nathalie; Ouedraogo, Moctar; Dailey-Chwalibog, Trenton; Hanley Cook, Giles; Ganaba, Rasmané; Lachat, Carl; Huybregts, Lieven; Kolsteren, Patrick

VERSION 1 – REVIEW

REVIEWER	Megan Bourassa New York Academy of Sciences, USA
REVIEW RETURNED	08-Apr-2020

GENERAL COMMENTS	This study protocol is well written and well thought out. The introduction provides some global and LMIC-specific interventions, but given that many nutrition interventions are context-specific, it would be nice to include some information on the previous studies done in Burkina Faso. There have been a number of recent studies on nutrient interventions during pregnancy in Burkina Faso, including multiple micronutrient supplements and lipid-based supplements. In this overview, I would also incorporate how the MISAME III study fits into the previous MISAME projects.
--

REVIEWER	Leila Nikniaz Tabriz health services management research center. Tabriz. University of Medical Sciences, Tabriz, Iran
REVIEW RETURNED	25-Jul-2020

GENERAL COMMENTS	I think this is a complete protocol for evaluating the effect of the supplement on birth outcomes. The question that arises is that do all the enrolled women have the same gestational age? Given that the energy supplementation affects pregnancy outcomes over time, the same duration of supplementation is essential for all women. professional language editing is needed.
---

REVIEWER	Hao, Liping Huazhong University of Science and Technology Tongji Medical College, Nutrition and Food Hygiene
REVIEW RETURNED	03-Nov-2020

GENERAL COMMENTS	This is a very meaningful study, but I still have some questions and suggestions about the study protocol.  1. Study population and recruitment : According to the study protocol, you intend to exclude pregnant women who are allergic to peanuts (page 8 Line 47). Why peanut? Should you consider excluding pregnant women with medical conditions that may affect the outcome of a pregnancy? 2. Considering the time effect on the association between nutrients during pregnancy and birth outcomes, it is necessary to intervene at the same pregnant stage. So, the intervention will start at which gestational week and end at which gestational week in this protocol? 3. The article mentioned that blinding of participants and community-based project workers will not be possible since the supplements are identifiable (page 10 line 46-47) and The FA will inform women on the supplement's function (page 8 line 60 and page 9 line 3). Will this operation cause a placebo effect? 4. What's the percent of total energy from protein of the fortified BEP supplements and what is your basis? 5. How to calculate dietary diversity score? 6. Sample size: Do you know the exclusive breastfeeding rate of the study population? It is recommended to consider the population's exclusive breastfeeding rate when determining the sample size of the postnatal intervention group.
--

VERSION 1 – AUTHOR RESPONSE

Reviewer: 1

Megan Bourassa

New York Academy of Sciences, USA

Please state any competing interests or state 'None declared':

None declared

Comments to the Author

Question 4:

This study protocol is well written and well thought out. The introduction provides some global and LMIC-specific interventions, but given that many nutrition interventions are context-specific, it would be nice to include some information on the previous studies done in Burkina Faso. There have been a number of recent studies on nutrient interventions during pregnancy in Burkina Faso, including multiple micronutrient supplements and lipid-based supplements. In this overview, I would also incorporate how the MISAME III study fits into the previous MISAME projects.

Answer from the Authors:

Added on page 4 and 5 of the Manuscript.

Reviewer: 2

Leila Nikniaz

Tabriz health services management research center. Tabriz. University of Medical Sciences, Tabriz, Iran

Please state any competing interests or state 'None declared':

No competing interests

Comments to the Author

Question 5 :

I think this is a complete protocol for evaluating the effect of the supplement on birth outcomes. The

question that arises is that do all the enrolled women have the same gestational age? Given that the energy supplementation affects pregnancy outcomes over time, the same duration of supplementation is essential for all women.

Answer :

The study aims at including pregnant women as early as possible. The initial screening relies on self-reported amenorrhea to a study community worker who visits every woman once every four weeks. Because of the RCT the main concern is to treat pregnant women in the intervention and control arm in a similar way. The study only includes women with a gestational age below 20 weeks to reduce the heterogeneity in gestational age at inclusion and to allow for sufficient time for the supplement to support pregnancy.

Question 6:

Professional language editing is needed.

Answer from the Authors:

Manuscript was checked by 2 native English speakers (see Manuscript for changes done with Track Changes).

Reviewer: 3

Hiping Lao

No competing interests

Comments to the Author

This is a very meaningful study, but I still have some questions and suggestions about the study protocol.

Question 7:

1. Study population and recruitment : According to the study protocol, you intend to exclude pregnant women who are allergic to peanuts (page 8 Line 47). Why peanut?

Should you consider excluding pregnant women with medical conditions that may affect the outcome of a pregnancy?

Answer :

Because the BEP supplement is peanut based. No other medical conditions are considered in our exclusion criteria.

Question 8 :

2. Considering the time effect on the association between nutrients during pregnancy and birth outcomes, it is necessary to intervene at the same pregnant stage. So, the intervention will start at which gestational week and end at which gestational week in this protocol?

Answer :

We aimed to include women as early as possible in pregnancy by announcing the start of our study in the villages, by visiting all eligible women in the area once a month and by adding the exclusion criterion of 20 weeks or more of gestational age for inclusion. The trial is an individually randomized controlled trial so gestational age at enrolment should be equally balanced between intervention and control.

The intervention will start as soon as possible (once a pregnancy is confirmed by a pregnancy test) and lasts up to delivery. All women randomised in the postnatal intervention group continue supplementation up to 6 months after birth.

Question 9 :

3. The article mentioned that blinding of participants and community-based project workers will not be possible since the supplements are identifiable (page 10 line 46-47) and The FA will inform women on the supplement's function (page 8 line 60 and page 9 line 3). Will this operation cause a placebo effect?

Answer from the Authors:

A placebo effect would appear if a group gets a supplement/medicine without an active compound which in this study, is not the case. Women belonging to the intervention group receive daily BEP supplement and iron folic acid tablets, while the control/comparison group receives iron and folic acid (standard of care). Iron and folic acid is expected to influence the main study outcome as well and can thus be seen as an active comparator.

Question 10 : 4.What's the percent of total energy from protein of the fortified BEP supplements and what is your basis?

Answer from the Authors:

A 72g sachet of BEP provides 393kcal and 14,5 grams protein. Twenty (20) % of total energy of the supplement comes from protein.

The compositional guidelines for macro- and micronutrients of the supplement as proposed by the Bill and Melinda Gates Foundation can be found on page 8 of the manuscript. The aim was to propose a composition of the supplement that could be widely use in different contexts and populations.

Ingredients of the Plumpy Mum distributed in MISAME-III (information from the Nutriset product sheet) : vegetable oils (rapeseed, palm, soy in varying proportions), sugar, whey powder, peanuts, defatted soy flour, maltodextrin (maize), soy protein concentrate, vitamin and mineral complex, stabilizer (fully hydrogenated vegetable fat, mono and diglycerides).

Question 11 : 5.How to calculate dietary diversity score?

Answer from the Authors:

Dietary diversity score will be measured by means of a twice weekly recall of the 10 food group indicator (1) by the trained village women.

(1) Yves Martin-Prevel and others, Moving Forward on Choosing a Standard Operational Indicator of Women's Dietary Diversity (Rome: FAO, 2015).

Question 12 : 6. Sample size: Do you know the exclusive breastfeeding rate of the study population?

It is recommended to consider the population's exclusive breastfeeding rate when determining the sample size of the postnatal intervention group.

Answer from the Authors:

It is not possible to find specific data on exclusive breastfeeding in that part of the country since there is a lot of variety across the country. According to data from the National Nutrition Survey in 2013, early and exclusive breastfeeding rates ranges from 20.5 percent to 74.1 percent.

The primary study outcome for the postnatal intervention is child growth. Exclusive breastfeeding is a secondary auxiliary study outcome for this study. We collect data on exclusive breastfeeding to verify if maternal BEP supplement is not associated with breastfeeding practices, but there is no study hypothesis ex ante. As such, we do not present a formal sample size calculation.

VERSION 2 – REVIEW

REVIEWER	Liping Hao School of Public Health, Tongji Medical College, Huazhong University of Science and Technology, Wuhan, China
REVIEW RETURNED	15-Jan-2021
GENERAL COMMENTS	I have no other comments, since authors have revised MS with great efforts.